# Efficient electron transmission in covalent organic framework nanosheets for highly active electrocatalytic carbon dioxide reduction

Hong-Jing Zhu[1,3], Meng Lu [1,3], Yi-Rong Wang[1], Su-Juan Yao[1], Mi Zhang[1], Yu-He Kan [2], Jiang Liu [1], Yifa Chen[1], Shun-Li Li[1] & Ya-Qian Lan [1]*

Efficient conversion of carbon dioxide ($CO_2$) into value-added products is essential for clean energy research. Design of stable, selective, and powerful electrocatalysts for $CO_2$ reduction reaction ($CO_2$RR) is highly desirable yet largely unmet. In this work, a series of metallo-porphyrin-tetrathiafulvalene based covalent organic frameworks (M-TTCOFs) are designed. Tetrathiafulvalene, serving as electron donor or carrier, can construct an oriented electron transmission pathway with metalloporphyrin. Thus-obtained M-TTCOFs can serve as electrocatalysts with high $FE_{CO}$ (91.3%, −0.7 V) and possess high cycling stability (>40 h). In addition, after exfoliation, the $FE_{CO}$ value of Co-TTCOF nanosheets (~5 nm) is higher than 90% in a wide potential range from −0.6 to −0.9 V and the maximum $FE_{CO}$ can reach up to almost 100% (99.7%, −0.8 V). The electrocatalytic $CO_2$RR mechanisms are discussed and revealed by density functional theory calculations. This work paves a new way in exploring porous crystalline materials in electrocatalytic $CO_2$RR.

[1] Jiangsu Collaborative Innovation Centre of Biomedical Functional Materials, Jiangsu Key Laboratory of New Power Batteries, School of Chemistry and Materials Science, Nanjing Normal University, Nanjing 210023, China. [2] Jiangsu Province Key Laboratory for Chemistry of Low-Dimensional Materials, School of Chemistry and Chemical Engineering, Huaiyin Normal University, Huai'an 223300, China. [3]These authors contributed equally: Hong-Jing Zhu, Meng Lu. *email: yqlan@njnu.edu.cn

Excessive utilization of fossil fuels and continuous human activity have led to the depletion of resources, energy crises, and global warming coupling with the pollution of high-level carbon dioxide ($CO_2$) (~411 p.p.m. in 2019)[1]. The yearly increased concentration of $CO_2$ has resulted in the rising of sea level, abnormal climate, ocean storms, and increased desertification area, etc[2]. To alleviate these problems, efficient conversion of $CO_2$ into high value-added products through methods such as electrochemical[3], photochemical[4], or thermochemical[5] approaches are essential and urgent. Among them, electrocatalytic $CO_2$ reduction reaction ($CO_2$RR) with the advantages of simple devices, high environmental compatibility, and the possibility of combination with other renewable energy sources (e.g., solar or wind energy) is considered as a kind of promising and alternative strategy[6,7]. However, owing to the inherent thermodynamic stability of $CO_2$ and competitive kinetically favored $H_2$ generation reaction, electrocatalytic $CO_2$RR generally faces drawbacks such as low reaction activity, selectivity, or electrical conductivity, which is far from meeting the demand of practical applications[8–10]. To conquer these problems, diverse electrocatalysts such as metals (e.g., Cu or Fe, etc.)[11,12], metal dichalcogenide (e.g., $WSe_2$, $Ag_2S$ or CuS, etc.)[13], and metal oxide (e.g., $Co_3O_4$, $Cu_2O$, or $SnO_2$, etc.)[14–16] have been explored for electrocatalytic $CO_2$RR. Yet, despite the intensive investigation, they still face some problems such as $CO_2$ adsorption or enrichment ability, intermolecular electron transmission efficiency, and electrocatalysis mechanism due to their non-porous or undefined structures.

Notably, porous crystalline materials such as covalent organic frameworks (COFs) with well-defined crystal structures have been explored as promising platforms in electrocatalytic $CO_2$RR[17–19]. COFs, a class of porous crystalline materials composed of light-weight elements and connected by strong covalent bonds, possess predictable structures, high stability, and porosity[20–23]. COFs serving as promising and alternative materials in electrocatalytic $CO_2$RR are mostly attributed to the following reasons: (i) compared with other materials with higher density, COFs with similar quality might provide more exposed surface area and active sites; (ii) the tunable structures endow COFs with diverse functionality such as electron donating, transferring, or $CO_2$ enrichment favorable for the enhancement of electrocatalytic $CO_2$RR performances; and (iii) various metal types (e.g., Co, Cu, and Ni, etc.) can be modified into their structures to impart COFs with tunable catalysis centers. Therefore, the syntheses of novel COFs and the exploration of them in electrocatalytic $CO_2$RR are very meaningful and highly demanded. However, despite the promising properties of COFs in electrocatalytic $CO_2$RR, only a few works about COFs (e.g., COF-366-Co and COF-367-Co, etc.) and their derivatives (e.g., modified with functional groups such as fluorine or methoxy) have been explored[17–19]. Nevertheless, the electrocatalytic $CO_2$RR efficiency of COFs are relatively low (the faradic efficiency of carbon monoxide ($FE_{CO}$), generally < 90%), which might be in part attributed to the low intermolecular electron transmission efficiency or lack in oriented electron transmission pathway. Besides, closely packed two-dimensional (2D) layer structure of COFs, especially in an eclipsed stacking fashion with strong $\pi$–$\pi$ interactions, will inevitably lead to insufficient utilization of the active sites and result in low electrocatalysis performances[24,25]. Exfoliation of the layered organic structure can expose larger surface area and more accessible active sites to facilitate the contact with substrate molecules, which would serve as an effective strategy to overcome this issue[26–28]. Nevertheless, the exploration of exfoliated 2D COF in electrocatalytic $CO_2$RR has not been reported. Therefore, the construction of novel COFs from functional units and further study the possibility of exfoliated COFs in efficient electrocatalytic $CO_2$RR are highly desirable.

Tetrathiafulvalene (TTF) as a kind of electron donor with high electron mobility is able to synthesize highly conductive charge-transfer crystals when constructed with electron acceptors[29,30]. A well-defined TTF-based structure might have the potential for the particular alignment and stacking of TTF columns as conductive pathways[31]. Metalloporphyrin, possessing conjugated $\pi$-electron system, can act as excellent electron acceptor and electron transfer carrier[3,32,33]. Combining TTF with metalloporphyrin might construct intermolecular charge-transfer pathway in a structure to largely enhance the electron transfer efficiency. Up to date, COFs that are based on only TTF or metalloporphyrin have been investigated[34–37]. The combination of them in a COF structure might be of high significance in electron transfer efficiency to enhance the electrocatalytic $CO_2$RR activity.

Herein, a series of stable and high crystalline metalloporphyrin-TTF based COFs (M-TTCOFs) are produced through the assembly of metallized 5,10,15,20-tetrakis (4-aminophenyl) porphinato (M-TAPP, M = Co or Ni) and 2,3,6,7-tetra (4-formylphenyl)-tetrathiafulvalene (4-formyl-TTF) (Fig. 1a). The synergistic combination of metalloporphyrin and TTF can serve as the role of gathering electron donating, electron migration, and electrocatalytic active components together in these M-TTCOFs. Combining high porosity, excellent chemical stability, and uniformly distributed metal centers in the structures, thus-obtained M-TTCOFs with diverse transition metals (i.e., Co or Ni) present superior electrocatalytic $CO_2$RR performances. Notably, Co-TTCOF is able to selectively convert $CO_2$ to CO with a $FE_{CO}$ of 91.3% at −0.7 V and possesses remarkable cycling stability (>40 h). Besides, after exfoliation, the $FE_{CO}$ of Co-TTCOF nanosheets (denoted Co-TTCOF NSs, ~5 nm in thickness) can reach up to 99.7% at −0.8 V, which is highest in reported COFs. Furthermore, the electrocatalytic $CO_2$RR mechanisms of M-TTCOFs with diverse metal centers imply that Co-TTCOF exhibits the lowest activation energy for the determine step in electrocatalytic $CO_2$RR compared with other M-TTCOFs as revealed by density functional theory (DFT) calculations, which can fully support the performances.

## Results

**Structure and characterization of M-TTCOFs.** The crystal structures of M-TTCOFs are resolved by using powder X-ray diffraction (PXRD) measurements in conjunction with Pawley refinements and the structural simulations are performed in Materials Studio 7.0. For example, taking Co-TTCOF, Co-TTCOF exhibits high crystallinity in the experimental PXRD test (Fig. 1b). The Pawley refinements reproduce the experimentally detected PXRD pattern with a negligible difference (Rp, 3.01% and Rwp, 4.34%), which indicates the correctness of the structure. In the PXRD pattern, the peak signals at 5.15° and 5.9° are assigned to the (110) and (200) facets, respectively (Supplementary Fig. 1). Meanwhile, Ni-TTCOF and $H_2$-TTCOF show intact topology with similar PXRD patterns as Co-TTCOF (Fig. 1b). In addition, Fourier-transform infrared spectroscopy and $^{13}C$ solid-state nuclear magnetic resonance ($^{13}C$ ssNMR) measurements are conducted to support the crystal structures of M-TTCOFs. Similar peak at 1622 cm$^{-1}$ in $H_2$-TTCOF and M-TTCOFs confirms the successful formation of C = N bond in the structures, accompanied by the diminish of C = O (1697 cm$^{-1}$) stretching band of 4-formyl-TTF and N–H (3324 cm$^{-1}$) and stretching band of 5,10,15,20-tetrakis (4-aminophenyl)-21H,23H-porphine ($H_2$-TAPP) (Supplementary Figs. 2 and 3). In $^{13}C$ ssNMR spectra of Co-TTCOF, the α-pyrrolic carbon present in the porphyrin moiety shows a peak at ~144.5 p.p.m. (peak b1). The dithiole carbon presented in the TTF shows a peak at ~113.8 p.p.m. (peak d1) and the peak around ~115.8 p.p.m. (peak d2) is attributed to β-pyrrolic

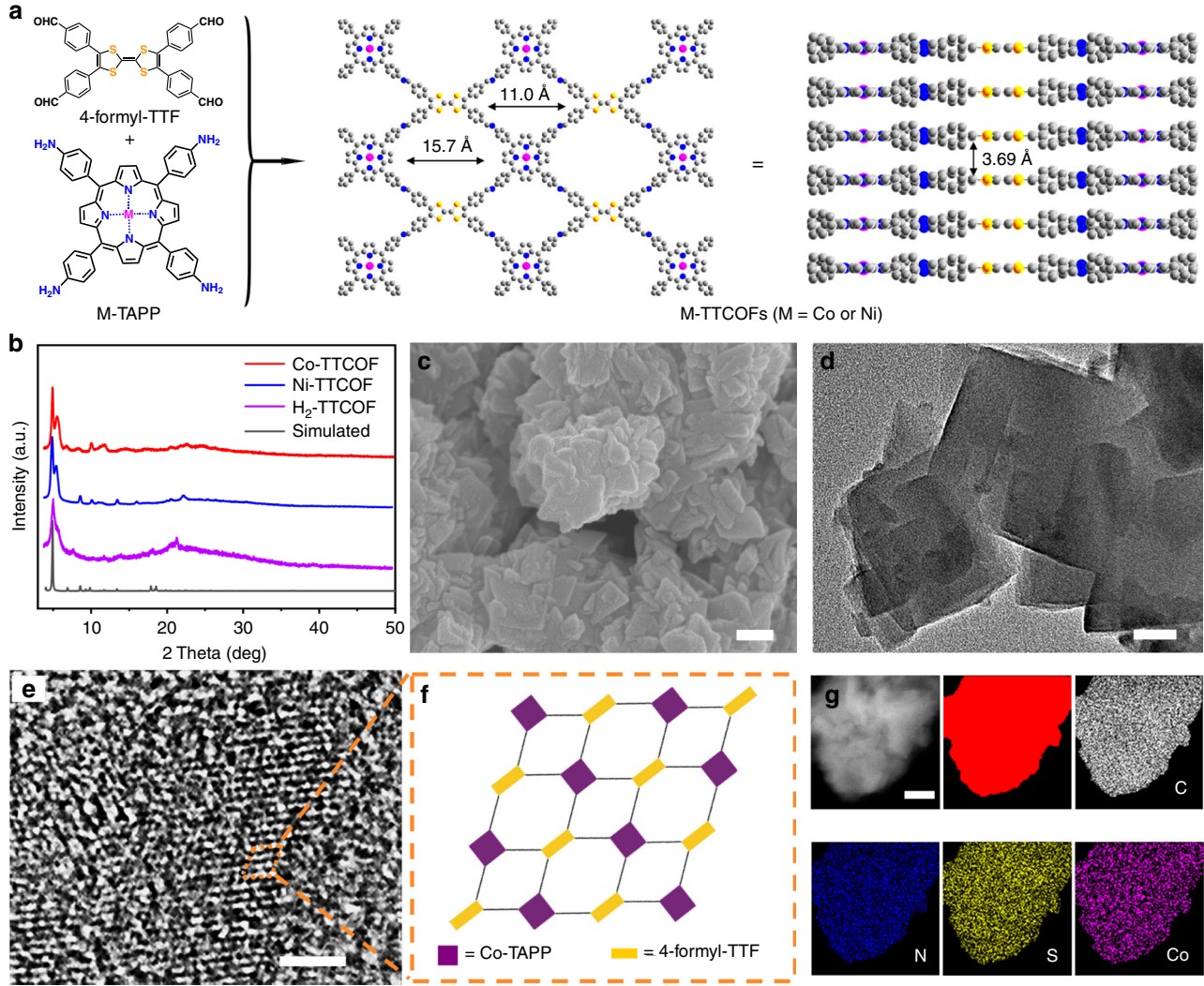

**Fig. 1 Structure and characterization of M-TTCOFs (M = Co or Ni). a** The structure of M-TTCOFs obtained through the condensation of 4-formyl-TTF and M-TAPP. **b** Powder X-ray diffraction patterns of M-TTCOFs. **c** Scanning electron microscopy image, scale bar = 200 nm. **d** Transmission electron microscopy image, scale bar = 50 nm. **e** High-resolution transmission electron microscopy image, scale bar = 2 nm. **f** The schematic pore structure of Co-TTCOF. **g** Energy-dispersive X-ray spectroscopy elemental mapping image (the red image is the selected area in **g**), scale bar = 100 nm.

carbon and methine carbon of the porphyrin macrocycle (Supplementary Fig. 4d). Besides, the peaks among the range from 124 to 136 p.p.m. (peak c) can reflect the existence of phenyl moiety in Co-TTCOF. Furthermore, the formation of C=N bond in Co-TTCOF is proved by the existence of resonance signal around 157.3 p.p.m. (peak a) (Supplementary Fig. 4)[33,38–40]. In addition, the solid-state UV test is performed to verify the structure of Co-TTCOF (Supplementary Fig. 5). The electronic absorption spectrum of Co-TTCOF displays a solid-state broad absorption giving four maxima (Q-band, 546, 593, and 738 nm; sorbet band 412 nm) in the visible region. The results indicate the existence of cobalt porphyrin unit in Co-TTCOF[41]. X-ray photoelectron spectroscopy (XPS) tests are further carried out to determine the surface electronic state and elemental composition of M-TTCOFs. The analyses show that the metal centers in M-TTCOFs (M = Co or Ni) are all bivalence (Supplementary Figs. 6 and 7)[42].

In the structure, the assembly of metalloporphyrin and TTF generates 2D layer porous structure, which might be beneficial for the mass transfer of substrates and enrichment of $CO_2$. To prove it, $N_2$ and $CO_2$ sorption tests are carried out. In $N_2$ sorption test, the surface area of Co-TTCOF is calculated to be 481 m$^2$ g$^{-1}$, which is

slightly lower than that of H$_2$-TTCOF (S$_{BET}$, 675 m$^2$ g$^{-1}$) without metal doping (Supplementary Figs. 8 and 9). The pore volume of Co-TTCOF (V$_t$, 0.633 cm$^3$ g$^{-1}$) is slightly increased compared with H$_2$-TTCOF (V$_t$, 0.612 cm$^3$ g$^{-1}$). For Ni-TTCOF, the surface area and pore volume are calculated to be 531 m$^2$ g$^{-1}$ and 0.483 cm$^3$ g$^{-1}$, respectively (Supplementary Fig. 10). Furthermore, $CO_2$ adsorption capacity of Co-TTCOF is measured to be 20 cm$^3$ g$^{-1}$ at 293 K (Supplementary Fig. 11). The value is higher than that of H$_2$-TTCOF (293 K, 11 cm$^3$ g$^{-1}$), which indicates that the doping of Co can enhance the adsorption capacity of $CO_2$. Similar results that metal doping can increase the adsorption capacity are also detected for Ni-TTCOF (293 K, 21 cm$^3$ g$^{-1}$) when compared with H$_2$-TTCOF (293 K, 11 cm$^3$ g$^{-1}$) (Supplementary Fig. 11).

The morphology of the M-TTCOFs are characterized by scanning electron microscopy (SEM) and transmission electron microscopy (TEM) tests. Taking Co-TTCOF for instance, SEM test shows that the morphology of Co-TTCOF is rectangular crystal with size about 150 nm, which is different from that of H$_2$-TTCOF (nanoparticle, about 100 nm) (Fig. 1c and Supplementary Fig. 12). The morphology is also supported by the TEM test (Fig. 1d). Besides, high-resolution TEM (HR-TEM) exhibits that Co-TTCOF displays highly ordered pore channels, which matches well with

the AA-stacking mode of the 2D Co-TTCOF (Fig. 1e, f)[43,44]. Energy-dispersive X-ray spectroscopy (EDS) mapping (Fig. 1g) analysis reveals that C, N, S, and Co are uniformly distributed in Co-TTCOF nanocrystals (Supplementary Fig. 13). The total Co content in Co-TTCOF is determined to be 3.4 wt% by inductively coupled plasma optical emission spectrometry (ICP-OES) (Supplementary Table 1). In comparison, the morphology of Ni-TTCOF is nanopellet with a size of about 1.4 μm proved by SEM and TEM tests (Supplementary Figs. 14 and 15).

Stability is a key factor to evaluate the durability of the catalysts in various applications[45]. To investigate the chemical stability of M-TTCOFs, the samples of M-TTCOFs are soaking in diverse solutions for 5 days. For example, taking Co-TTCOF, the internal structure of Co-TTCOF remains intact after immersing in boiling water and 0.5 M $KHCO_3$ (pH 7.2) for 5 days (Supplementary Fig. 16). Moreover, Co-TTCOF can be stable in 0.1 M HCl (pH 1) and 0.1 M KOH (pH 13) for more than 5 days (Supplementary Fig. 16). Similar results are also detected for Ni-TTCOF (Supplementary Fig. 17). Besides, TGA analyses under $N_2$ atmosphere are conducted to evaluate the stability of these COFs (i.e., $H_2$-TTCOF, Co-TTCOF, and Ni-TTCOF). The results indicate that these COFs can be stable up to 350 °C (Supplementary Fig. 18). The high chemical and thermal stability of M-TTCOFs sets fundamental basis for further applications in electrocatalytic $CO_2$RR.

**The electrocatalytic performances of M-TTCOFs.** Above all, the specially designed M-TTCOFs constructed from TTF (electron donating unit) and metalloporphyrin (electron accepter or electron transfer unit) with highly porous structures, excellent chemical stability, and uniformly distributed metal centers might serve as promising candidates for electrocatalytic $CO_2$RR. To test their performances, M-TTCOFs (i.e., Co-TTCOF, Ni-TTCOF, and $H_2$-TTCOF) are packaged in cells and tested in three-electrode electrochemical H-type cell with $CO_2$ or Ar-saturated 0.5 M $KHCO_3$ solution as the electrolyte. In this work, all potentials are measured using Ag/AgCl electrode as the reference electrode and the results are reported relative to the reversible hydrogen electrode (RHE).

Linear sweep voltammetry (LSV) curves (without iR compensation) show that the onset potential of Co-TTCOF (−0.45 V) is much more positive than that of Ni-TTCOF (−0.64 V) in $KHCO_3$ solution (Fig. 2a). Tests conducted both in Ar and $CO_2$ saturated $KHCO_3$ solution show that Co-TTCOF exhibits a higher current density in $CO_2$-saturated $KHCO_3$ solution than that in Ar-saturated $KHCO_3$ solution over a wide potential range (−0.3 to −1.0 V vs. RHE), which suggests higher reaction activity of electrocatalytic $CO_2$RR than HER (Supplementary Fig. 19). Furthermore, the gas chromatography (GC) analysis show that $H_2$ and CO are the primary reduction products and there is no liquid product detected by [1]H NMR spectroscopy (Supplementary Fig. 20). Besides, the Tafel slope of Co-TTCOF is tested to be 237 mV dec$^{-1}$, which is much smaller than that of Ni-TTCOF (629 mV dec$^{-1}$) and $H_2$-TTCOF (433 mV dec$^{-1}$) (Fig. 2b). This implies the more favorable kinetics of Co-TTCOF in generation of CO, which might be ascribed to the more efficient charge-transfer ability and larger active surface during the reaction process.

To probe the electrocatalytic kinetics on the electrode/electrolyte surface, electrochemical impedance spectroscopy (EIS) measurement is carried out. Interestingly, the Nyquist plots demonstrate that Co-TTCOF has much smaller charge-transfer resistance (23.4 Ω) than Ni-TTCOF (191.5 Ω) and $H_2$-TTCOF (111.9 Ω) during the electrocatalytic $CO_2$RR (Supplementary Fig. 21). It implies faster electron transfer from the catalyst surface to the reactant (i.e., $CO_2$) in intermediate (HCOO* and CO*) generation, thus eventually resulting in largely enhanced activity and selectivity for Co-TTCOF. To estimate the electrochemical active surface area (ECSA) and

further to discuss the potential influence factors, electrochemical double-layer capacitance ($C_{dl}$) is calculated (Supplementary Fig. 22). The obtained results show that Co-TTCOF presents a $C_{dl}$ value of 6.00 mF cm$^{-2}$, which is higher than Ni-TTCOF (3.87 mF cm$^{-2}$) and $H_2$-TTCOF (4.27 mF cm$^{-2}$). To calculate the percent of electrochemically active cobalt, cyclic voltammetry (CV) tests of Co-TTCOF and Ni-TTCOF are conducted. Peak current and scan rate as two important parameters are detected to reveal the percent of electrochemically active sites. The peak current curve shows a linear dependence on the scan rate (tested from 20 to 120 mV s$^{-1}$) for Co-TTCOF (Supplementary Fig. 23). Regression of the linear regime between 20 and 100 mV s$^{-1}$ with equation: slope = $n^2F^2A\tau_o/4$ RT gives the surface concentrations ($\tau_o$) of electroactive Co-TTCOF to be $7.05 \times 10^{-9}$ [17,41]. Based on the results, the percent of electrochemically active cobalt is calculated to be 0.90% for Co-TTCOF.

To further confirm the structure superiority in electrocatalytic $CO_2$RR of the obtained M-TTTCOFs over other COF structures such as COF-366-$H_2$ (constructed from 1,4-benzenedicarboxaldehyde and metalloporphyrin), relatively direct current conductivity tests are performed (Supplementary Fig. 24)[17]. Specially, $H_2$-TTCOF presents larger slope (1/R) value than COF-366-$H_2$, which indicates the synergistic effect of metalloporphyrin and TTF in creating oriented electron pathway. In this work, metallated porphyrin is mostly possible to be an electron acceptor when coupling with strong donating component such as TTF. To prove it, we have performed related CV and optical tests to reveal the relative energy levels (i.e., lowest unoccupied molecular orbital (LUMO) and highest occupied molecular orbital (HOMO)) of monomers and presented a detailed discussion about donor–acceptor concept (Supplementary Figs. 25–27). The HOMO levels are calculated from the onset of the first oxidation waves (i.e., 5,10,15,20-tetrakis (para-aminophenyl) porphyrin Cobalt (II) (Co-TAPP), $E^{OX} = 0.69$ V and 4-formyl-TTF, $E^{OX} = 0.48$ V) from CV tests. The band gaps estimated from Tauc plot of solid-state UV show the Eg (band gaps) for Co-TAPP and 4-formyl-TTF are 1.70 and 1.63 eV, respectively (Supplementary Fig. 26b, d). The relative positions of LUMO and HOMO are obtained according to the formula (HOMO = $-[(eE^{OX}-eE(Fc/Fc^+) + 4.8$ V)] eV, LUMO = HOMO − Eg). Based on first oxidation waves data, 4-formyl-TTF is a better electron donor as indicated by its lower oxidation potential (0.48 V vs. Ag/AgCl in $CH_3$CN) than Co-TAPP (0.69 V vs. Ag/AgCl in $CH_3$CN). Further supported by the LUMO levels of Co-TAPP and 4-formyl-TTF, in which the LUMO of 4-formyl-TTF possesses higher potential (−3.30 eV) than Co-TAPP (−3.14 eV), which is sufficient to realize the electron transfer from 4-formyl-TTF to Co-TAPP (Supplementary Fig. 27). To further prove it, the comparison of Co-TAPP and Co-TTCOF in XPS tests reveals an apparent change of binding energy, which also provides a direct evidence that the charge carrier migration pathway might be from TTF to Co-TAPP (Supplementary Fig. 28)[46].

To determine the carbon source of CO, an isotopic experiment that using $^{13}CO_2$ as substrate is performed under identical reaction conditions. The products are analyzed by GC and mass spectra. As presented in Fig. 2f, the peak at $m/z = 29$ is assigned to $^{13}$CO, demonstrating that the carbon source of CO indeed derives from the $CO_2$ used. Under the condition that Ar-saturated $KHCO_3$ solution applied as the electrolyte, only $H_2$ is detected by the GC (Supplementary Fig. 29). Besides, the bare carbon cloth and carbon cloth decorated with acetylene black and Nafion are measured as comparisons and no electrocatalytic $CO_2$RR activity are detected (Supplementary Figs. 30 and 31).

Moreover, corresponding $FE_{CO}$ and $FE_{H_2}$ are calculated over the entire potential range to further evaluate the selectivity of the M-TTCOFs for electrocatalytic $CO_2$RR (Fig. 2c). Taking

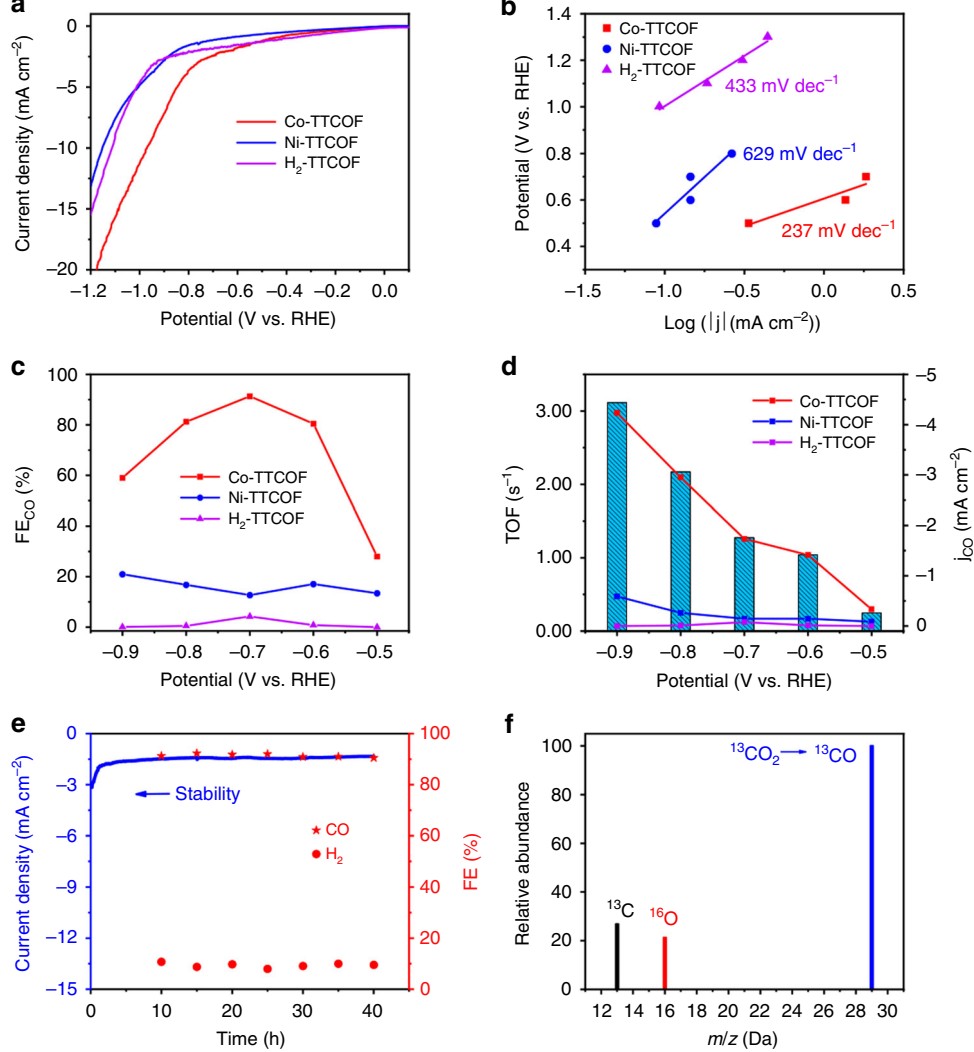

**Fig. 2 Electrocatalytic performances of M-TTCOFs. a** Linear sweep voltammetry curves. **b** Tafel plots. **c** The faradic efficiency of carbon monoxide calculated overpotential range from −0.5 to −0.9 V. **d** Partial CO current density and TOF(s⁻¹). **e** Cycling stability test of Co-TTCOF at the potential of −0.7 V *vs*. RHE. **f** The mass spectra test of Co-TTCOF of $^{13}CO$ recorded under $^{13}CO_2$ atmosphere.

Co-TTCOF for instance, the initial formation of CO is detected by GC with a CO partial current density of 0.10 mA cm⁻² at the potential of −0.45 V. With the increase of potential, the $FE_{CO}$ continuously enhances and reaches up to a maximum value of 91.3% at −0.7 V (Supplementary Fig. 32 and Eqs. 1 and 2). The maximum $FE_{CO}$ value of Co-TTCOF (91.3%) is higher than other M-TTCOFs (i.e., Ni-TTCOF, 20.9% and H₂-TTCOF, 4.22%). Specially, the $FE_{CO}$ of Co-TTCOF is ~32 and 4 times higher than those of H₂-TTCOF and Ni-TTCOF, respectively. Besides, Co-TAPP as one of the precursors is tested as comparison and it exhibits a $FE_{CO}$ of 69% at −0.9 V (Supplementary Fig. 33). Noteworthy, the best performance of Co-TTCOF is superior to reported COFs (e.g., COF-366-Co ($FE_{CO}$, 90%, −0.55 V), COF-366-F-Co ($FE_{CO}$, 87%, −0.55 V), and COF-300-AR ($FE_{CO}$, 80%, −0.55 V)) in electrocatalytic $CO_2RR$ (Supplementary Table 2)[17–19]. To further support the remarkable performances of M-TTCOFs, partial current densities of CO and H₂ at different potentials are detected (Fig. 2d and Supplementary Figs. 34 and 35). Co-TTCOF gives a partial CO current density of 1.84 mA cm⁻² at −0.7 V. This value is more than ten times larger than those of Ni-TTCOF (0.15 mA cm⁻²) and H₂-TTCOF (0.079 mA cm⁻²). Besides, the turnover frequency (TOF) of Co-TTCOF is calculated to be 1.28 s⁻¹ at −0.7 V.

Long-time durability is an important parameter to estimate the performance in electrocatalytic $CO_2RR$, as it determines the lifetime of electrocatalysts. To study it, the $CO_2RR$ stability of Co-TTCOF is performed with chronoamperometric test at a fixed potential of −0.7 V in 0.5 M KHCO₃ solution. After 40 h, negligible decay in activity is detected (the gaseous product is analyzed by GC every 10 h) (Fig. 2e). During the process, the corresponding $FE_{CO}$ can be retained at values >90% over the entire experiment, which implies Co-TTCOF to be a highly stable electrocatalyst. Based on the results, the TON of Co-TTCOF is calculated for this process. Notably, the TON (CO) of Co-TTCOF is as high as 40,142 in just 10 h and can reach up to 141,479 after 40 h. Besides, the TON (H₂) of Co-TTCOF is 4014 in 10 h and can reach up to 14,148 after 40 h (Supplementary Fig. 36). To evaluate the durability of Co-TTCOF, HR-TEM, SEM, and XPS tests of Co-TTCOF after electrocatalysis are performed. The SEM and TEM images of Co-TTCOF agree well with the state before electrocatalysis, indicating Co-TTCOF can maintain its morphology after electrocatalysis (Supplementary Fig. 37). Besides, the XPS tests after electrocatalysis show that the valence state of Co (II) remains almost unchanged (i.e., Co2p₃/₂, 780.86 eV and Co2p₁/₂, 760.16 eV) when compared with that of Co-TTCOF (i.e., Co2p₃/₂, 780.84 eV and Co2p₁/₂, 796.14 eV) before electrocatalysis

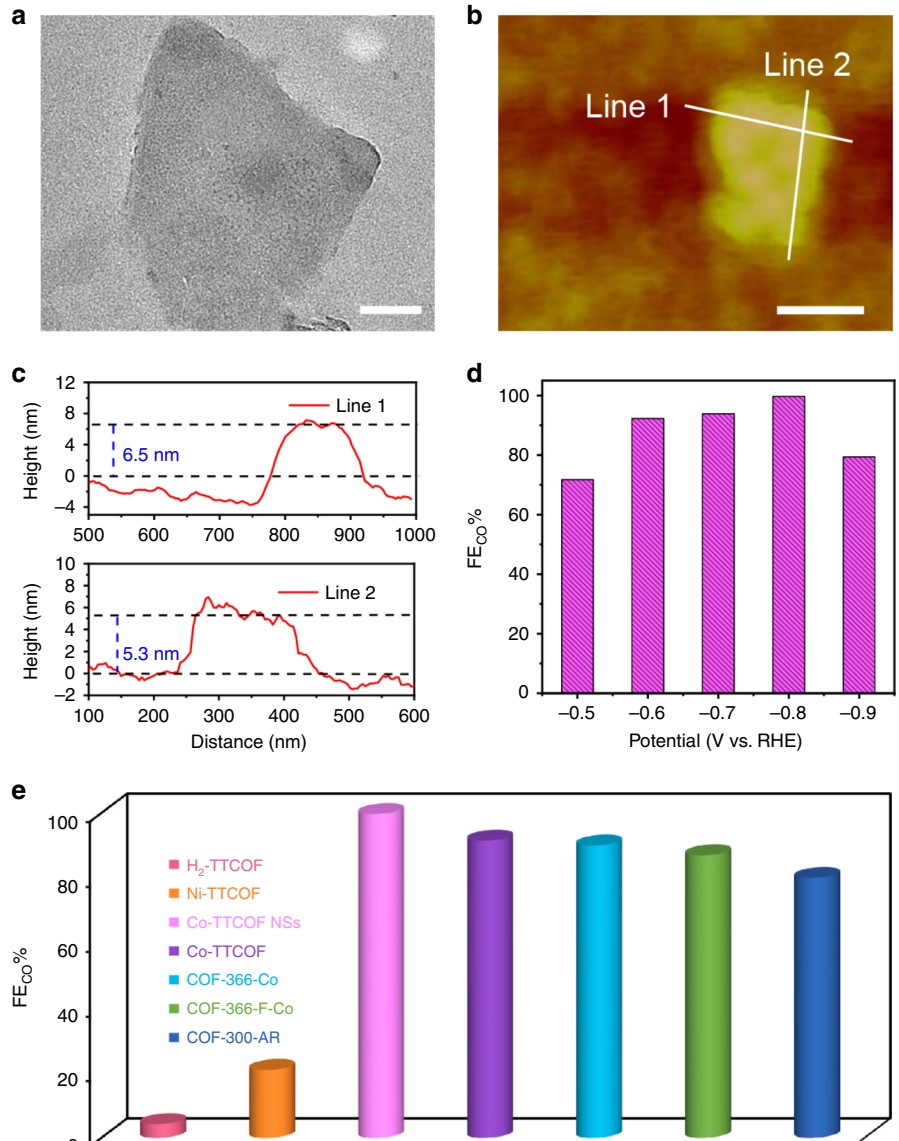

**Fig. 3 Characterization and electrocatalytic performances of Co-TTCOF NSs. a** TEM image, scale bar = 100 nm. **b** The atomic force microscope topographical image, scale bar = 100 nm. **c** Height profile (line 1 and 2 represent the lines in **b**. **d** $FE_{CO}$ of Co-TTCOF nanosheets over a potential range from −0.5 to −0.9 V. **e** A summary of electrocatalytic performances of the literature reported materials and M-TTCOFs.

(Supplementary Fig. 38). Moreover, ICP test of Co-TTCOF after long-time durability test (>40 h) has been conducted and negligible leaching of metal ions are detected in the electrolyte. These results indicate Co-TTCOF to be excellent electrocatalyst with high durability, which might be attributed to the strong covalent bonds generated in COFs.

In addition, we have tested the pH dependence of overpotential values[41,47]. After numerous trial-and-error processes, acidic electrolyte with pH values of 4.8, 5.8, and 6.8 are picked as three representative ones to investigate the pH dependence of overpotential values (Supplementary Fig. 39). LSV tests (without $iR$ compensation) and electrocatalytic $CO_2RR$ performances of Co-TTCOF as two kinds of powerful methods are applied to reveal the pH dependence of overpotential values from different aspects. In LSV curves, the overpotential at 1 mA cm$^{-2}$ decreases from ~410 mV (pH 4.8) to ~340 mV (pH 5.8) and finally slightly increases to ~350 mV (pH 6.8) with the increase of pH values (Supplementary Figs. 39a, b). Although for the electrocatalytic performances of Co-TTCOF, the overpotential (based on the

highest $FE_{CO}$%) of Co-TTCOF reaches to ~790 mV both for pH 4.8 and pH 5.8, then the value decreases to ~590 mV (pH 6.8). The results of these two methods indicate that the overpotential value is closely related to the pH of electrolyte.

Based on the remarkable stability and high performances of M-TTCOFs, methods that can further increase the performances of M-TTCOFs are meaningful to explore the tunability and applicability in practical applications of these porous materials. As a kind of 2D material, the potential exfoliation ability of M-TTCOFs in converting bulks into nanosheets might have improvements in the performance of electrocatalytic $CO_2RR$. Herein, we report the first case of exfoliated COF in the application of electrocatalytic $CO_2RR$. By high-frequency sonication for 30 min, the original bulk crystals are transformed into Co-TTCOF NSs with a size of ~200 nm proved by TEM test (Fig. 3a). After exfoliation, the inert structure of Co-TTCOF remains intact in the nanosheets as certified by PXRD tests (Supplementary Fig. 40). To evaluate the thickness of the obtained Co-TTCOF NSs, atomic force microscope (AFM) tests

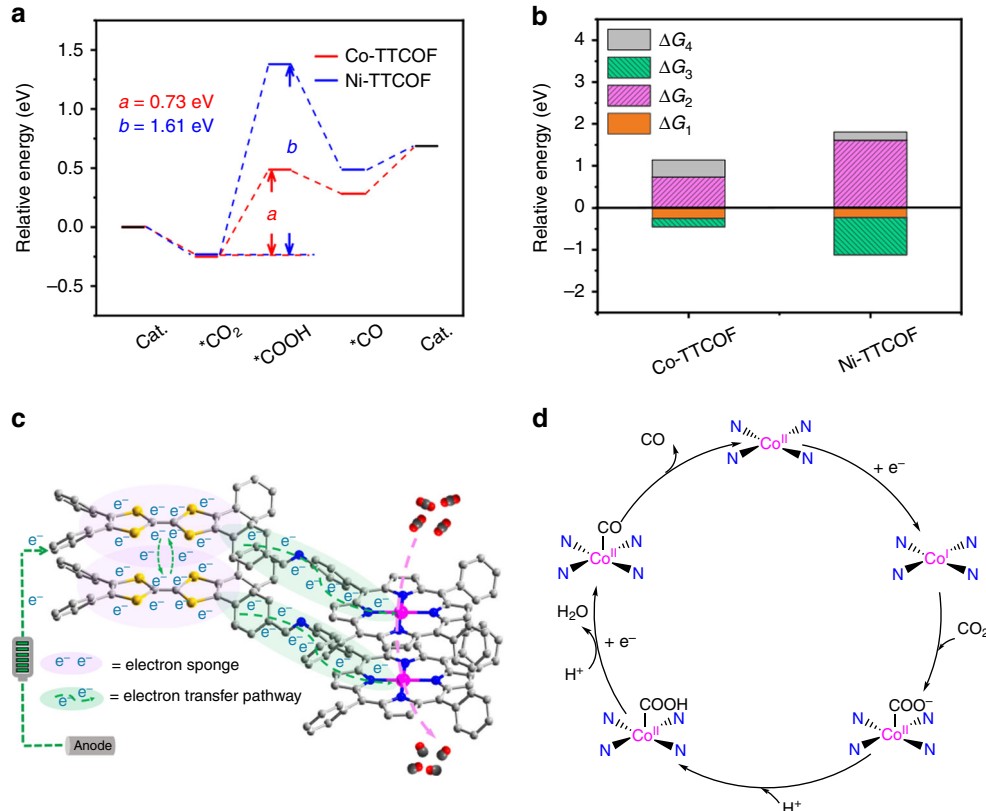

**Fig. 4 DFT calculations and proposed schematic mechanism of M-TTCOFs. a** The relative energy diagrams of $CO_2$ reduction to CO for M-TTCOFs (M = Co or Ni). **b** Comparison of the relative energy of each elementary reaction ($\Delta G_1$, $\Delta G_2$, and $\Delta G_3$ represent the free energy of *$CO_2$, *COOH, and *CO formation, respectively, and $\Delta G_4$ stands for the free energy of CO desorption process) in electrocatalytic $CO_2$RR for M-TTCOFs (M = Co or Ni). **c, d** Proposed schematic mechanism for the electrocatalytic $CO_2$RR on Co-TTCOF.

are conducted. The results detected in different regions of the nanosheets show that the thickness of Co-TTCOF NSs ranges from 5 to 6 nm (Fig. 3b, c). To further evaluate the selectivity of the Co-TTCOF NSs for electrocatalytic $CO_2$RR, the corresponding $FE_{CO}$ and $FE_{H_2}$ are calculated over the entire potential range. Remarkably, the $FE_{CO}$ value is higher than 90% in a wide potential range of −0.6 to −0.9 V (Fig. 3d). The maximum $FE_{CO}$ value ($FE_{CO}$, 99.7%) of Co-TTCOF NSs at −0.8 V is higher than unexfoliated one ($FE_{CO}$, 91.3%). Noteworthy, the $FE_{CO}$ value is also highest among reported COFs (Fig. 3e)[17–19].

**The DFT calculation and reaction mechanism**. To understand the high activity and reaction mechanism of Co-TTCOF, the DFT calculations are performed (Supplementary Note 1). In general, the electroreduction of $CO_2$ to CO includes three elementary reactions, the formation of *COOH and *CO with one electron transfer for each of them and the CO desorption process (Fig. 4a). The asterisk (*) represents the surface active sites for adsorption and reaction. When assembling TTF and metalloporphyrin together, Co-TTCOF possesses remarkably reduced $\Delta G_2$ (free energy for *COOH formation in the rate-determining steps (RDS), 0.73 eV) and $\Delta G_3$ (free energy for *CO formation, −0.20 eV) compared with other M-TTCOFs, being consistent with its higher electroreduction activity and selectivity (Fig. 4a, b). For Ni-TTCOF, the RDS for the formation of *COOH become harder ($\Delta G_2$, Ni-TTCOF, 1.61 eV) when compared with Co-TTCOF (Fig. 4a, b). In a word, the results of the reaction energies for M-TTCOFs with different metal centers can fully support the experiments, which can further confirm the high activity of the Co-TTCOF.

Based on the experiment results and theoretical calculations, possible reduction processes and mechanism from $CO_2$ to CO on Co-TTCOF are elaborated (Fig. 4c, d). During electrocatalytic $CO_2$RR, Co centers in Co-TTCOF are reduced from Co(II) to Co (I) proved by the cyclic voltammetry (CV) tests, which has been detected in many works (Supplementary Fig. 41)[3,17]. As presented in the schematic diagram, TTF as a kind of electron sponge and donator with high electron mobility can construct efficient electron transmission pathway with metalloporphyrin (Fig. 4c). During the electrocatalytic $CO_2$RR process, TTF initially traps the electron from the electrode and efficiently transfers it to Co center through the electron transmission pathway (Fig. 4c). Meanwhile, the Co center is reduced from Co(II) to Co(I) and then Co(I) interacts with carbon monoxide to give Co(II)*COOH intermediate (Fig. 4d). Finally, Co(II)*COOH converts to Co(II) *CO and CO is desorbed (Supplementary Note 2).

## Discussion

In summary, we have designed and synthesized a series of stable COFs via Schiff-base condensation reaction of metalloporphyrin and TTF. The synergistic combination of TTF and metalloporphyrin in these M-TTCOFs can serve as the role of gathering electron donating, electron migration, and electrocatalytic active components together in the electrocatalytic $CO_2$RR. Imparting with high porosity, excellent chemical stability and uniformly distributed metal centers, thus-obtained M-TTCOFs present excellent electrocatalytic $CO_2$RR performances. Remarkably, Co-TTCOF is able to selectively convert $CO_2$ into CO with a $FE_{CO}$ of 91.3% at −0.7 V. Notably, after exfoliation, the $FE_{CO}$ of Co-TTCOF nanosheets (~5 nm in thickness) is higher than 90% in a

wide potential range from $-0.6$ to $-0.9$ V and the maximum $FE_{CO}$ can reach up to almost 100% (99.7%, $-0.8$ V), which might be attributed to the larger surface area and more accessible active sites. Furthermore, the electrocatalytic $CO_2RR$ mechanisms of M-TTCOFs with diverse metal centers show that Co-TTCOF has the lowest activation energy for the determine step in electrocatalytic $CO_2RR$ compared with other M-TTCOFs, which matches well with their performances. This strategy opens great perspectives in designing novel and efficient $CO_2RR$ electrocatalysts to address $CO_2$ problems.

## Materials and methods

**Synthesis of 2,3,6,7-tetra (4-formylphenyl) tetrathiafulvalene**. The synthesis method of 4-formyl-TTF follows previously reported procedure with a slight modification[48,49]. In detail, 4-formyl-TTF (1.0 g), 4-bromobenzaldehyde (4.5 g), Pd (OAc)$_2$ (0.28 g), PtBu$_3$HBF$_4$ (1.0 g), and Cs$_2$CO$_3$ (5.9 g) were added in a 100 mL three-neck round-bottomed flask. After purification with high-purity nitrogen gas for three times, anhydrous tetrahydrofuram (THF) (50 mL) was added under nitrogen atmosphere. After that, the flask was heated at 75 °C to reflux and kept stirring for 30 h. After cooling to room temperature, the organic compounds were extracted with CHCl$_3$ (100 mL) for three times to collect the organic phase (dark red) followed with vacuum filtration to remove the undissolved solid. Then, the obtained sample was washed with brine (100 mL) for three times by using a pear-shaped separatory funnel and was dried with anhydrous Na$_2$SO$_4$. After that, the organic phase was collected to afford crude product (dark red) through rotary evaporation to remove the solvent. The resulted sample was purified by column chromatography with silica using dichloromethane:acetone = 500:3 as mobile phase to achieve pure product 4-formyl-TTF (0.9 g) (red, ~30% yield).

**Synthesis of 5,10,15,20-tetrakis (para-aminophenyl) porphyrin cobalt (II)**. The synthesis method of Co-TAPP follows previously reported procedures[17]. In detail, H$_2$-TAPP (200 mg, 0.3 mmol) and Co(OAc)$_2$·2H$_2$O (212 mg, 1.2 mmol) were added in a 250 mL three-neck round-bottomed flask. After purification with high-purity nitrogen gas for three times, a solvent mixture of methanol (20 mL), chloroform (90 mL), and N,N'-dimethylformamide (30 mL) were added. The flask was heated at 80 °C under stirring and nitrogen atmosphere for 24 h. After cooling to room temperature, the solution was transferred into a separatory funnel and washed with water (3 × 100 mL). After that, the solution was collected through rotary evaporation to afford dark purple solid (Co-TAPP, ~180 mg, ~85% yield).

**Synthesis of Co-TTCOF**. The synthesis method of Co-TTCOF follows previously reported procedures[33]. Co-TAPP (14.7 mg, 0.02 mmol), 4-formyl-TTF (12.4 mg, 0.02 mmol), 1, 4-dioxane (0.5 mL), 1,3,5-trimethylbenzene (0.5 mL), and 6 M aqueous acetic acid (0.2 mL) were mixed in a Pyrex tube (outside diameter × length, 19 × 65 mm). After sonication for about 15 min, the tube was flash frozen at 77 K (liquid N$_2$ bath) and degassed to achieve an internal pressure of ~100 mTorr. After the temperature recovers to room temperature, the mixture was heated at 120 °C and left undisturbed for 72 h. After filtration, the wet sample was transferred to a Soxhlet extractor and washed with THF (24 h) and acetone (24 h). Finally, the product was evacuated at 150 °C under dynamic vacuum overnight to yield activated sample (28.4 mg, ~80% yield based on Co-TAPP).

**Synthesis of H$_2$/Ni-TTCOF**. The syntheses of H$_2$/Ni-TTCOF followed similar procedures as Co-TTCOF, except that Co-TAPP was replaced with H$_2$-TAPP (13.5 mg, 0.02 mmol) and Ni-TAPP (14.7 mg, 0.02 mmol), respectively.

**Synthesis of Co-TTCOF NSs**. Co-TTCOF bulks (30 mg) were added into a beaker and then high-frequency (1000 W) sonication was carried out for 30 min with deionized water (100 mL) at room temperature (25 °C). After filtration (filter paper, ~220 nm), the obtained sample was collected and dried under vacuum at 60 °C for 12 h.

**Characterizations and instruments**. PXRD patterns were recorded on a D/max 2500VL/PC diffractometer (Japan) equipped with a graphite monochromatized Cu Kα radiation source ($\lambda$ = 1.54060 Å). The corresponding working voltage and current are 40 kV and 100 mA, respectively. TEM and HR-TEM images were recorded on JEOL-2100F apparatus at an accelerating voltage of 200 kV. Morphological and microstructural analyses were conducted using a SEM (JSM-7600F) at an accelerating voltage of 10 kV. EDS was performed with a JSM-5160LV-Vantage type energy spectrometer. Nitrogen adsorption–desorption isotherms were recorded at 77 K using a Quantachrome instrument (Quantachrome Instruments Autosorb IQ2). XPS was performed on a scanning X-ray microprobe (PHI 5000 Verasa, ULAC-PHI, Inc.) using Al Kα radiation and the C 1s peak at 284.8 eV as the internal standard. ICP-OES (Leeman Labs) was used to measure the content of metal ions. Hydrogen amounts were analyzed using a gas chromatograph (GC-7900, CEAULight, China) equipped with a thermal conductivity detector (TCD).

AFM was performed on tapping mode (Vecco Nanoscope Multimode 8.0). Converting Co-TTCOF bulks into nanosheets used ultrasonic cell grinder (VOSHIN-1000W). The relatively direct current conductivity tests were conducted with a probe station at room temperature (25 °C) under ambient conditions with a computer-controlled analog-to-digital converter (2636B, Kethley). The conductive sample was pressed into a sheet using a ton of pressure in the mold and the test voltage was among the range from $-200$ to 300 mV.

**Electrochemical measurements**. All electrocatalysis tests of the catalysts were performed at ambient environment on the electrochemical workstation (SP-150, Bio-Logic) in a standard three-electrode configuration in 0.5 M KHCO$_3$. Carbon rod and Ag/AgCl electrode were used as the counter and reference electrode, respectively. The experiment was performed in an airtight electrochemical H-type cell with a catalyst-modified carbon cloth electrode (denoted as CCE, 1 cm × 2 cm) as the work electrode.

Given the poor intrinsic electrical conductivity of M-TTCOFs, acetylene black was introduced to mix with M-TTCOFs to improve the conductivity. Nafion solution was introduced as a kind of electrocatalyst dispersion solution, which can form a homogeneous ink with COF and further help to attach onto the surface of carbon cloth. The preparation procedures of the CCE working electrode was presented as follows: 10 mg electrocatalyst, 3 mg acetylene black, and 1 mL 0.5% Nafion solution were grounded to form uniform catalyst ink. After sonication for 30 min, the ink was dropped directly onto a carbon cloth (1 cm × 1 cm) with a catalyst loading density of ~1 mg cm$^{-2}$ and dried. In the H-type cell, two compartments were separated by an exchange membrane (Nafion®212). We have measured the thicknesses of working electrodes before and after the deposition of COF materials using a vernier caliper (all the samples are properly seized on the vernier caliper without deformation), but there are no obviously change (~0.30 mm) in the macroscopic range (Supplementary Fig. 42a, b). Therefore, we intend to detect the thickness of the working electrode from microscopic perspective through SEM test. For example, taking Co-TTCOF based working electrode, the average diameter of sample-coated carbon cloth is about 9.8 μm compared with the bare carbon cloth (7.8 μm). Based on the result, the thickness of the coating is estimated to be about 1 μm (Supplementary Fig. 42c, d).

During the electrocatalytic $CO_2RR$ experiments, the polarization curves were performed by LSV mode at a scan rate of 2 mV s$^{-1}$. Initially, polarization curves for the modified electrode were recorded under an inert N$_2$ atmosphere. After that, the solution was bubbled with $CO_2$ (99.999%) for at least 30 min to make the aqueous solution saturated and then the electrocatalytic $CO_2RR$ test was conducted. Potential was measured vs. Ag/AgCl electrode and the results were reported vs. RHE based on the Nernst equation: $E$ (vs. RHE) = $E$ (vs. Ag/AgCl) + 0.1989 V + 0.059 × pH.

EIS spectroscopy measurement was carried out by applying an AC voltage with 10 mV amplitude in a frequency range from 1000 kHz to 100 mHz at overpotential of $-0.7$ V (vs. RHE). To estimate the ECSA, CV were tested by measuring $C_{dl}$ under the potential window range from 0.05 V to 0.25 V (vs. RHE) with various scan rates from 10 to 100 mV s$^{-1}$. All the LSV curves were presented without $iR$ compensation.

**Chemicals and materials**. All solvents and reagents obtained from commercial sources were used without further purification. TTF (≥98%) was purchased from J&K China Chemical Ltd. Mesitylene (≥99.5%) was purchased from Sinopharm Chemical Reagent Co., Ltd. H$_2$-TAPP was purchased from Kaiyulin (Shanghai) Development Co., Ltd. 1, 4-Dioxane (≥99.7%) was purchased from MACKLIN reagent. 4-Bromobenzaldehyde was purchased from Meryer (Shanghai) Chemical Technology Co., Ltd. Exchange membrane (Nafion®212) Cs$_2$CO$_3$ was purchased from Tokyo Chemical Industry (Shanghai) Co., Ltd. CHCl$_3$ and Na$_2$SO$_4$ were purchased from China National Medicines Corporation Ltd.

**Reaction product analysis and calculation**. The electrolysis was carried out in an airtight electrochemical H-type cell at selected potentials ($-0.5$ to $-0.9$ V) to determine the reduction products and their Faradic efficiency. The gaseous reduction products (e.g., CO) were monitored by a GC (GC-7920) equipped with a flame ionization detector. During the test, nitrogen was the carrier gas. A TCD was used to analyze hydrogen with nitrogen as the carrier gas.

The liquid products were collected from the cathode chambers after electrolysis and quantified by NMR (Bruker AVANCEAV III 400) spectroscopy, in which 0.5 mL electrolyte was mixed with 0.1 mL D$_2$O. Solvent pre-saturation technique was implemented to suppress the water peak.

The calculation of Faradaic efficiency

For CO,

$$FE = \frac{2F \times n_{co}}{i \times t} \times 100\% \tag{1}$$

For H$_2$,

$$FE = \frac{2F \times n_{H_2}}{i \times t} \times 100\% \tag{2}$$

where F is the Faraday constant, $n_{CO}$ is the moles of produced CO, $t$ is the time (s), and $n_{H_2}$ is the moles of produced H$_2$.

Turnover frequency (TOF, $s^{-1}$)
The TOF for CO was calculated as follows:

$$TOF = \frac{i \times E_F}{N \times F \times n_{tot}} \qquad (3)$$

The TON for CO was calculated as follows:

$$TON = \frac{Q \times E_F}{N \times F \times n_{tot}} \qquad (4)$$

where $Q$ is the total charge passed in time, $i$ is the current, $E_F$ is the Faradaic efficiency for the desired product, $N$ is the number of electrons in the half reaction ($N = 2$ for $CO_2$ to CO conversion), $F$ is the Faraday constant ($F = 96485$ C $mol^{-1}$ electrons), and $n_{tot}$ is the total moles of catalyst employed in the electrolysis. The TON is calculated on the basis of the actually catalytic activity.

Regression of the linear regime between 20 and 120 mV $s^{-1}$ with equation gives the surface concentrations:

$$Slope = \frac{n^2 F^2 A \tau_0}{4RT} \qquad (5)$$

where $\tau_0$ is the surface concentrations, $n$ is number of electrons involved, $R$ is gas constant, and $T$ is temperature (298 K).

## Data availability
The data that support the findings of this study are available from the corresponding author upon reasonable request. The source data underlying Figs. 1b–e, 1g, 2a-f, 3a-e, and 4a-b are provided as a Source Data file.

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

## Acknowledgements

This work was financially supported by NSFC (Numbers 21622104, 21701085, 21871141, and 21871142); the NSF of Jiangsu Province of China (Number BK20171032); the Natural Science Research of Jiangsu Higher Education Institutions of China (Number 17KJB150025); Priority Academic Program Development of Jiangsu Higher Education Institutions and the Foundation of Jiangsu Collaborative Innovation Center of Biomedical Functional Materials.

## Author contributions

Y.-Q.L., S.-L.L., H.-J.Z., and M.L. conceived the idea. H.-J.Z., M.L., J.L., and Y.C. designed the experiments, collected, and analyzed the data. Y.-R.W. and M.Z. assisted with the experiments and characterizations. S.-J.Y. and Y.-H.K. accomplished the theoretical calculation. H.-J.Z. wrote the manuscript. All authors discussed the results and commented on the manuscript.

## Competing interests

The authors declare no competing interests.
