## [Peer Review File · Nature Communications]

Reviewers' comments:

Reviewer #1 (Remarks to the Author):

Recommendation: Major revisions needed as noted.

Ms. No.: NCOMMS-19-26978

Title: 'Efficient Electron Transmission in Covalent Organic Framework Nanosheets for Highly Active Electrocatalytic CO₂ Reduction'

Authors: Hong Jing Zhu, Meng Lu, Yi Rong Wang, Su-Juan Yao, Mi Zhang, Yu He Kan, Jiang Liu, Yifa Chen, Shun Li Li and Ya Qian Lan.

Comments:

In this manuscript Zhu et al. reported the synthesis of electroactive metal-porphyrin-Tetrathiafulvalene based Covalent Organic Framework and study the electrocatalytic activity for CO₂ reduction under basic conditions. On electrocatalytic CO₂ reduction can get published in Nature Communications after substantial more work has been performed. In particular control experiments/a fair comparison to other systems are missing, which makes it really hard to estimate how well the system performs. Also the extremely high metal loading is a concern which needs to be addressed. Another concern is the lack of details in the experimental section, which will make it very hard for others to reproduce these results. Overall the research is interesting and suitable for the readership of Nature Communications. However, further revision is needed to make this work publishable in Nature Communications.

The authors should clarify the following points.

1. Donor-acceptor concept is not suitable for this type of COF. Tetrathiafulvalene and metallated porphyrin both can act as electron donors. The authors should explain this in detail- what they meant for donor-acceptor polymer in this case and relative energetics by measuring electrochemistry of monomers.
2. For the xps, do they observe any shift in the binding energy of cobalt, Nickel once they are in COF structures in comparison to cobalt porphyrin monomer? The same thing goes for N as well.
3. What is the thickness of the working electrode after deposition of COF materials?
4. What percent of cobalt or Nickel porphyrin is electrochemically active?
5. They should also perform the catalysis with cobalt porphyrin only and explain the advantages of using it in this COFs support,
6. The authors should show the pH dependence of overpotential values.
7. The authors can also perform solid state UV to confirm the presence of cobalt porphyrin.
8. The authors can also perform TGA analysis for all the COFs materials.
9. Does cobalt or Nickel leach into the solution after the run? Again this needs to be measured via ICP (post experiment measurement).
10. Manuscript writing should be improved; the authors have failed to cite recent references in introduction part, In the introduction: ACS Catal. 2017, 7, 6120, ACS Appl. Mater. Interfaces, 2019, 11, 1520 and ACS Appl. Mater. Interfaces 2017, 9, 23843 are relevant under COF, CMP catalysis with metal porphyrin CMPs, COFs should be cited.
11. Electrocatalytic properties of COFs, which composed of free-base porphyrin, should be checked to verify the key role of Co²⁺ and Ni²⁺ as the active site.
12. To established durability of your catalyst, you should mention material characterization data like HRTEM, FE SEM, XPS after the CO₂ reduction experiment (post characterisation data).
13. The efficacy of the catalyst is well represented by TON and this value should be given. What is TON for this process?
14. The NMR is too broad and unresolved. Deconvolution of the solid state ¹³C NMR spectra would be needed to state this COFs structure (e.g. Macromolecules, 2018, 51, 3088).

Reviewer #2 (Remarks to the Author):

In this work, the authors reported a series of metalloporphyrin-tetrathiafulvalene based covalent organic frameworks (M-TTCOFs). Tetrathiafulvalene, serving as electron donor or carrier, can construct oriented electron transmission pathway with metalloporphyrin. The obtained M-TTCOFs can be used as electrocatalysts with high FE_{CO} (91.3%, -0.7 V) and possess remarkable cycling stability (> 40 h) in electrocatalytic CO_2RR . After exfoliation, the FE_{CO} of COF nanosheets (~5 nm) is higher than 90% in a wide potential range from -0.6 to -0.9 V and the maximum FE_{CO} can reach up to almost 100% (99.7%, -0.8 V), which have been largely improved compared with bulk material. The electrocatalytic CO_2RR mechanisms are discussed and revealed by DFT calculations. The data presented can fully support the conclusion. This work is quite novel and important, and paves a new way in exploring porous crystalline materials in electrocatalytic CO_2RR . Therefore, I recommend the publication in Nature Communications after minor revisions.

I would like to suggest the followings to the authors:

1. What is the main concern of choosing tetrathiafulvalene as the desired electron donor to construct COFs? It would be nice to have some discussion in the introduction, from the viewpoint of structure design or target properties.
2. The detailed procedure about the test of the relatively direct current conductivity should be presented, such as pre-treatment of the samples and experimental operation. Besides, some discussion about this test should be provided.
3. In order to exclude the influence of substrate and additives, the carbon cloth with acetylene black and Nafion would be measured in CO_2 -saturated 0.5 M $KHCO_3$ aqueous solution.
4. Please provide the TON of Co-TTCOF for the production of H_2 and CO.
5. The authors mentioned the performances of diverse electrocatalysts (e.g., metals, metal dichalcogenide and metal oxide) in the introduction. It would be better to list their CO_2RR performances in Supplementary file.
6. The exfoliated COF was produced by high-frequency sonication, while the specific work parameters of ultrasound such as frequency and temperature were not listed. Please add it.
7. Metalloporphyrin or tetrathiafulvalene based COFs are a class of promising materials for electrocatalysis. Some important works of metalloporphyrinic or tetrathiafulvalenic COFs (e.g., *Chem. Sci.*, **2014**, *5*, 4693; *Chem. Eur. J.* **2014**, *20*, 14614; *Angew. Chem. Int. Ed.*, **2019**, *58*, 6430, etc.) might be better cited to further support the novelty of the specially designed metalloporphyrin-tetrathiafulvalene based COFs.
8. The description of the caption in the Supplementary Figure 23 and Supplementary Figure 25 are incorrect. For example, "Ar-saturated" in "the FE_{CO} and FE_{H_2} of pure carbon cloth at different applied potentials in Ar-saturated 0.5 M $KHCO_3$ aqueous solution" is " CO_2 -saturated".

Responses to the Reviewers' Comments

Reviewer 1:

Comments to the Author:

In this manuscript Zhu et al. reported the synthesis of electroactive metal-porphyrin-Tetrathiafulvalene based Covalent Organic Framework and study the electrocatalytic activity for CO₂ reduction under basic conditions. On electrocatalytic CO₂ reduction can get published in Nature Communications after substantial more work has been performed. In particular control experiments/a fair comparison to other systems are missing, which makes it really hard to estimate how well the system performs. Also the extremely high metal loading is a concern which needs to be addressed. Another concern is the lack of details in the experimental section, which will make it very hard for others to reproduce these results. Overall the research is interesting and suitable for the readership of Nature Communications. However, further revision is needed to make this work publishable in Nature Communications. The authors should clarify the following points.

1. Donor-acceptor concept is not suitable for this type of COF. Tetrathiafulvalene and metallated porphyrin both can act as electron donors. The authors should explain this in detail what they meant for donor-acceptor polymer in this case and relative energetics by measuring electrochemistry of monomers.

Response:

Thanks for your kind suggestion. Tetrathiafulvalene (TTF) is mostly regarded as electron donor (refs. *Chem. Eur. J.* **2013**, 19, 338; *Chem. Soc. Rev.* **2018**, 47, 5614) and metallated porphyrin has been reported as either electron donor (ref. *J. Am. Chem. Soc.* **2008**, 130, 9451) or acceptor (refs. *Nat. Commun.* **2018**, 9, 4466; *J. Phys. Chem. C* **2014**, 118, 13503). In this work, metallated porphyrin is mostly possible to be electron acceptor when coupling with strong donating component like tetrathiafulvalene. To prove it, we have performed related cyclic voltammogram and optical tests to reveal the relative energy levels (i.e. LUMO and HOMO) of monomers and presented a detailed discussion about donor-acceptor concept (Supplementary Figs. 25-27). The HOMO levels are estimated from the onset of the first oxidation waves (Co-TAPP, $E^{\text{OX}} = 0.69$ V and 4-formyl-TTF, $E^{\text{OX}} = 0.48$ V) from cyclic voltammogram tests (refs. *Chem. Mater.* **1998**, 10, 30; *Chem. Eur. J.* **2014**, 20, 14614). The band gaps calculated from Tauc plot of solid state UV show the E_g (band gaps) for Co-TAPP and 4-formyl-TTF are 1.70 and 1.63 eV, respectively (refs: *J. Mater. Chem. A*, **2013**, 1, 4358; *Chem. Eur. J.* **2013**, 19, 338). The relative positions of LUMO and HOMO are obtained according to the formula ($\text{HOMO} = -[(eE^{\text{OX}} - eE(\text{Fc}/\text{Fc}^+) + 4.8 \text{ V})]$ eV, $\text{LUMO} = \text{HOMO} - E_g$). Based on first oxidation waves data, 4-formyl-TTF is better electron donor as revealed by its lower oxidation potential (0.48 V vs. Ag/AgCl in CH₃CN) than Co-TAPP (0.69 V vs. Ag/AgCl in CH₃CN). Further supported by the LUMO levels of Co-TAPP and 4-formyl-TTF, in which the LUMO of 4-formyl-TTF possesses higher potential (-3.30 eV) than Co-TAPP (-3.14 eV), which is sufficient to realize the electron transfer from 4-formyl-TTF to

Co-TAPP.

Based on these experimental results, donor-acceptor concept might be suitable for this type of COFs. Metalloporphyrin and TTF are connected to construct metalloporphyrin-TTF based COFs. In the structure, TTF is a sulfur-rich conjugated molecule with two reversible and easily accessible oxidation states (i.e., radical TTF⁺ cation and TTF²⁺ dication) that have been widely studied as a kind of electron donor (ref. *J. Am. Chem. Soc.* **2012**, 134, 12932). Co-porphyrin with inherent macrocycle conjugated π -electron system is very beneficial for electron mobility and Co(II) enables to be reduced to Co(I) during the process in many references (refs. *Nat. Commun.* **2018**, 9, 4466; *Science* **2015**, 349, 1208). The connection of tetrathiafulvalene and metalloporphyrin will presumably create an oriented electron transmission pathway under the motivation of electric field and thus obtained COF based materials exhibits excellent CO₂RR performances (FE_{CO}% = 91.3, -0.7V; long-time durability, 40 h).

Now we have added relative discussion in the revised manuscript (page 7, line 39; page 3, line 3) and Supporting Information (page 26, Supplementary Fig. 26; page 27, Supplementary Fig. 27).

Supplementary Figure 25. Cyclic voltammograms of ferrocene (internal standard) in 0.1 M nBu₄NPF₆ in CH₃CN at room temperature ($E(\text{Fc}/\text{Fc}^+) = 0.35$ V).

Supplementary Figure 26. The cyclic voltammogram and optical tests of Co-TAPP and 4-formyl-TTF. **a** Cyclic voltammograms of Co-TAPP. **b** Solid state UV of Co-TAPP (inset Tauc plot). **c** Cyclic voltammograms of 4-formyl-TTF. **d** Solid state UV of 4-formyl-TTF (inset Tauc plot).

Supplementary Figure 27. LUMO (red) and HOMO (black) levels of Co-TAPP and 4-formyl-TTF.

2. For the xps, do they observe any shift in the binding energy of cobalt, Nickel once they are in COF structures in comparison to cobalt porphyrin monomer? The same thing goes for N as well.

Response:

Thanks for your insightful suggestion. We have observed the shift in the binding energy of cobalt, nickel and N in COF structures compared with metallated porphyrin monomer. Co-TTCOF/Co-TAPP and Ni-TTCOF/Ni-TAPP have been selected as target pairs to demonstrate the possible binding energy shift in XPS spectra. The spectra of Co-TTCOF and Co-TAPP display one pair of peaks arising from the spin-orbit doublet of Co2p, which can be assigned to the Co2p_{3/2} and Co2p_{1/2} (Supplementary Fig. 28 a, b). The Co2p_{3/2} and Co2p_{1/2} peaks of Co-TTCOF locate at 780.84 eV and 796.14 eV, which presents apparent positive shift of ~0.25 eV and ~0.24 eV compared with pristine Co-TAPP (Co2p_{3/2}, 780.59 eV and Co2p_{1/2}, 795.90 eV). Besides, the binding energy of N1s for Co-TTCOF (Co-N bond, 398.64 eV) displays a negative shift of ~0.1 eV when compared with Co-TAPP (Co-N bond, 398.54 eV). Similar phenomenon is also detected for Ni-TTCOF, in which 0.07 and 0.06 eV shift are observed for Ni2p_{3/2} and Ni2p_{1/2} when compared with Ni-TAPP (Supplementary Fig. 28c, d). Also for N1s, a 0.03 eV shift is detected for Ni-TTCOF in contrast to Ni-TAPP. The change of binding energy provides direct evidence that the charge carrier migration pathway might be from TTF to M-TAPP (M = Co, Ni) as supported by the HOMO-LUMO results in Response 1 and are also verified by other important works (refs. *Adv. Mater.* **2019**, 31, 1802981; *Nat. Commun.* **2018**, 9, 1425; *Angew. Chem. Int. Ed.* 10.1002/anie.201907826) (Figure R1-1 to 1-3).

Now we have added relative references (ref. 46) and discussion in the revised manuscript (page 8, line 6) and Supporting Information (page 28, line 5, Supplementary Fig. 28).

Supplementary Figure 28. High-resolution XPS spectrum of Co-TTCOF and Ni-TTCOF. **a** Co2p for Co-TTCOF. **b** N1s for Co-TTCOF. **c** Ni2p for Ni-TTCOF. **d** N1s for Ni-TTCOF.

Figure below reproduced with permissions from Low et al., *In Situ Irradiated X-ray Photoelectron Spectroscopy Investigation on a Direct Z-Scheme TiO₂/CdS Composite Film Photocatalyst*, *Adv. Mater.* 2019. Copyright © 1999-2019 John Wiley & Sons, Inc. All rights reserved

Figure R1-1. High-resolution XPS for Ti 2p (b) and Cd 3d (c) of TiO₂/CdS in the dark or under 365 nm LED irradiation (UV-TiO₂/CdS) (*Adv. Mater.* 2019, 31, 1802981). In this paper, upon light irradiation, there was a slight positive shift (by 0.3 eV) in the Ti 2p binding energy, suggesting a decrease in its electron density under light irradiation. Meanwhile, two characteristic peaks attributed to CdS at Cd 3d_{5/2} and Cd 3d_{3/2} were observed without light, which underwent negative shift (by -0.2 eV) under light irradiation, suggesting an increase in the electron density on the CdS. These binding energy shifts provide direct evidence of the charge carrier migration pathway across the TiO₂/CdS interface. In detail, the photogenerated electrons migrate from TiO₂ to CdS.

Figure below reproduced with permissions from Wu et al., *Electron density modulation of NiCo₂S₄ nanowires by nitrogen incorporation for highly efficient hydrogen evolution catalysis*, *Nat. Comm.* 2018

Figure R1-2. XPS and XANES characterization of NiCo₂S₄ and N-NiCo₂S₄. XPS core-level spectra of **a** Co2p, **c** Ni2p, and **e** S2p for NiCo₂S₄ (blue) and N-NiCo₂S₄

(red). XANES spectra of **b** Co L-edge, **d** Ni L-edge, and **f** S L-edge for NiCo₂S₄ (blue) and N-NiCo₂S₄ (red) (ref. *Nat. Commun.* **2018**, 9, 1425). Figure R1-2 shows the XPS Co 2p core-level spectra of NiCo₂S₄ and N-NiCo₂S₄ NWs. Both spectra display one pair of peaks arising from the spin-orbit doublet of Co 2p, which can be assigned to Co 2p_{3/2} and Co 2p_{1/2}. The binding energies of Co 2p_{3/2} and Co 2p_{1/2} display a positive shift of ~0.4 eV after nitrogen incorporation. The electrons of the metal atoms prefer to flow towards nitrogen atoms.

Figure below reproduced with permissions from Yang et al., *Donor-Acceptor Nanocarbon Ensembles to Boost Metal-Free All-pH Hydrogen Evolution Catalysis by Combined Surface and Dual Electronic Modulation*, *Angew. Chem. Int. Ed* 2019. Copyright © 1999-2019 John Wiley & Sons, Inc. All rights reserved

Figure R1-3. High resolution N1s XPS spectra of MHCf and MHCf-CNTs (ref. *Angew. Chem. Int. Ed.* 10.1002/anie.201907826). As exhibited in the Figure R1-3, pyridinic, pyrrolic, quaternary N peaks for MHCf-CNTs shift to lower binding energy by 0.2, 0.3 and 0.3 eV respectively, signifying the existence of the electron transfer from CNTs to MHCf, resulting in the electron enrichment over MHCf.

3. What is the thickness of the working electrode after deposition of COF materials?

Response:

Thanks for your suggestion. We have measured the thicknesses of working electrodes before and after the deposition of COF materials using a vernier caliper, but there is no obviously change (~0.30 mm) in the macroscopic range. Therefore, we intend to detect the thickness of the working electrode from microscopic perspective through SEM test. Taking Co-TTCOF based working electrode for example, the average diameter of sample coated carbon cloth is about 9.8 μm compared with the bare carbon cloth (7.8 μm). Based on the result, the thickness of the coating is estimated to be about 1 μm (Supplementary Fig. 42). This might be ascribed to the special fabrication procedure of working electrode: M-TTCOFs (10 mg) was well-mixed with acetylene black (3 mg) and 0.5% Nafion solution (1 mL) to form a homogeneous ink. Then the mixture was uniformly dropped onto a carbon cloth (1 cm × 1 cm) and the working electrode was obtained after drying.

Now we have added relative discussion in the revised manuscript (page 14, line 26) and Supporting Information (page 42, Supplementary Fig. 42).

Supplementary Figure 42. The thickness test using a vernier caliper and SEM images. **a** The thickness test of carbon cloth using a vernier caliper (all the samples are properly seized on the vernier caliper without deformation). **b** The thickness test of working electrode using a vernier caliper. **c** The SEM image of carbon cloth. **d** The SEM image of working electrode.

4. What percent of cobalt or Nickel porphyrin is electrochemically active?

Response:

Thanks for your insightful suggestion. To calculate the percent of electrochemically active cobalt or nickel, cyclic voltammogram tests of Co-TTCOF and Ni-TTCOF are conducted. Peak current and scan rate as two important parameters are detected to reveal the electrochemically active sites. The peak current shows a linear dependence on the scan rate (tested from 20 to 120 mV s^{-1}) both for Co-TTCOF and Ni-TTCOF (Supplementary Fig. 23, refs. *Nat. Commun.* **2018**, 9, 4466; *Chem. Commun.* **2019**, 55, 11634). Regression of the linear regime between 20 and 120 mV s^{-1} with equation: slope = $n^2 F^2 A \tau_0 / 4 R T$ (n = number of electrons involved; F = Faraday constant in C mol^{-1} ; A = geometrical surface area of the electrode (0.071 cm^2); τ_0 = surface coverage; R = gas constant; T = temperature (298 K)) gives the surface concentrations (τ_0) of electroactive Co-TTCOF and Ni-TTCOF to be 7.05×10^{-9} and $1.50 \times 10^{-9} \text{ mol cm}^{-2}$ (ref. *ACS Appl. Mater. Interfaces.* **2019**, 11, 1520). Based on the results, the percent of electrochemically active cobalt or nickel are calculated to be 0.90% and 0.19% for Co-TTCOF and Ni-TTCOF, respectively.

Now we have added relative reference (ref. 41) and discussion in the revised manuscript (page 7, line 23) and Supporting Information (page 23, Supplementary Fig. 23).

Supplementary Figure 23. The cyclic voltammogram tests of Co-TTCOF and Ni-TTCOF.

5. They should also perform the catalysis with cobalt porphyrin only and explain the advantages of using it in this COFs support.

Response:

Thanks for your kind suggestion. According to your suggestion, the catalysis performance of cobalt porphyrin has been evaluated as comparison. Linear sweep voltammetry (LSV) curves (without iR compensation) show that the onset potential of Co-TTCOF (-0.45 V) is much more positive than that of Co-TAPP (-0.51 V) in CO_2 -saturated KHCO_3 solution (Supplementary Fig. 33a). Furthermore, Co-TAPP exhibits a FE_{CO} of 69% at -0.9 V, which is inferior to Co-TTCOF (91.3%, -0.7 V) (Supplementary Fig. 33b, d). Besides, Co-TAPP gives a partial CO current density of 0.48 mA cm^{-2} , which is much less than that of Co-TTCOF (1.84 mA cm^{-2}) at -0.7 V (Supplementary Fig. 33c). As shown in Response 1, Co-TAPP (0.69 V vs. Ag/AgCl in CH_3CN) is electron acceptor as indicated by its higher oxidation potential than 4-formyl-TTF (0.48 V vs. Ag/AgCl in CH_3CN) (Supplementary Fig. 26). In this work, Co-porphyrin and TTF are connected by robust covalent bonds to construct Co-TTCOF. Co-porphyrin with inherent macrocycle conjugated π -electron system is very beneficial for electron mobility and Co(II) enables to be reduced to Co(I) during the process as revealed in many references (refs. *Nat. Commun.* **2018**, 9, 4466;

Science **2015**, 349, 1208). Thus-obtained Co-TTCOF exhibits excellent CO₂RR performances (FE_{CO}% = 91.3, -0.7 V; long-time durability, 40 h), which can further verify the advantages of using cobalt porphyrin in this COFs system.

Now we have added relative discussion in the revised manuscript (page 8, line 27; page 7, line 39) and Supporting Information (page 33, line 3, Supplementary Fig. 33; page 26, Supplementary Fig. 26).

Supplementary Figure 33. Electrochemical performances of Co-TAPP. **a** LSV curves. **b** Faradaic efficiencies for CO. **c** Partial current density for CO. **d** FE_{CO} calculated over potential range from -0.5 to -0.9 V.

Supplementary Figure 26. The cyclic voltammogram and optical tests of Co-TAPP and 4-formyl-TTF. **a** Cyclic voltammograms of Co-TAPP. **b** Solid state UV of Co-TAPP (inset Tauc plot). **c** Cyclic voltammograms of 4-formyl-TTF. **d** Solid state UV of 4-formyl-TTF (inset Tauc plot).

6. The authors should show the pH dependence of overpotential values.

Response:

Per suggestion, we have tested the pH dependence of overpotential values. To test it, electrolyte with various pH values (from 4.8 to 13.8) are applied and tested. An interesting phenomenon is observed that the alkaline solutions (pH = 7.8-13.8) are acidified by the saturated CO₂ and the final pH values are reduced to ~6.8. After numerous trial-and-error processes, acidic electrolyte with pH values of 4.8, 5.8 and 6.8 are picked as three representative ones to investigate the pH dependence of overpotential values (Supplementary Fig. 39, ref. *Chem. Sci.* **2016**, 7, 1521). Linear sweep voltammetry tests (LSV, without *iR* compensation) (ref. *ACS Appl. Mater. Interfaces* **2019**, 11, 1520) and electrocatalytic CO₂RR performances of Co-TTCOF (ref. *Nat. Energy.* **2019**, 4, 732) as two kinds of powerful methods are applied to reveal the pH dependence of overpotential values from different aspects. In LSV curves, the overpotential at 1 mA cm⁻² decreases from ~410 mV (pH, 4.8) to ~340 mV (pH, 5.8) and finally slightly increases to ~350 mV (pH, 6.8) with the increase of pH values (Supplementary Figs. 39a, b). While for the electrocatalytic performances of Co-TTCOF, the overpotential (based on the highest FE_{CO}%) of Co-TTCOF reaches to ~790 mV both for pH = 4.8 and pH = 5.8, then the value decreases to ~590 mV (pH, 6.8). The results of these two methods indicate that the overpotential value is closely related to the pH of electrolyte (Supplementary Figs. 39c, d).

Now we have added relative references (refs. 41 and 47) and discussion in the revised manuscript (page 9, line 8) and Supporting Information (page 39, line 4, Supplementary Fig. 39).

Supplementary Figure 39. The pH dependence of overpotential values for Co-TTCOF. **a** LSV curves. **b** Overpotential ($\eta @ 1 \text{ mA cm}^{-2}$) vs pH plot. **c** FE_{CO} calculated over potential range from -0.5 to -0.9 V in various solution. **d** Overpotential ($\eta @ \text{highest FE}_{\text{CO}}\%$) vs pH plot.

7. The authors can also perform solid state UV to confirm the presence of cobalt porphyrin.

Response:

Thanks for your kind suggestion. According to your suggestion, the solid state UV tests of Co-TTCOF is performed. The electronic absorption spectrum of Co-TTCOF displays a solid-state broad absorption giving four maxima (Q-band, 546, 593 and 738 nm; sorbet band 412 nm) in the visible region. The results indicate the existence of cobalt porphyrin unit in Co-TTCOF (ref. *ACS Appl. Mater. Interfaces*. **2019**, 11, 1520).

Now we have added relative references (ref. 41) and discussion in the revised manuscript (page 5, line 4) and Supporting Information (page 5, Supplementary Fig. 5).

Supplementary Figure 5. The solid state UV spectra of Co-TTCOF.

8. The authors can also perform TGA analysis for all the COFs materials.

Response:

Per suggestion, we have performed TGA analyses under N_2 atmosphere for all of the COFs (i.e. H_2 -TTCOF, Co-TTCOF and Ni-TTCOF). Taking Co-TTCOF for example, the TGA plot shows a slight weight loss of 6.3%, which might be attributed to the loss of solvent (Supplementary Fig. 18a). Then the material remains stable up to ~ 350 °C, followed by a sharply weight loss ($\sim 74.6\%$) until 550 °C and finally the plateau remains stable without weight loss up to 800 °C. The remaining mass is about 14.0 wt%, which might be ascribed to the formation of cobalt oxide. Similar result is detected for Ni-TTCOF (Supplementary Fig. 18b, there plateaus: 4.9%, 63.8 % and 27.7%). For H_2 -TTCOF without metal doping, about 3.7% mass loss at temperature range from 25 to 200 °C is attributed to the loss of guest molecules. After 350 °C, the framework of H_2 -TTCOF starts to collapse and ends at about 600 °C (Supplementary Fig. 18c).

Now we have added relative discussion in the revised manuscript (page 5, line 49) and Supporting Information (page 18, Supplementary Fig. 18).

Supplementary Figure 18. TGA analyses of M-TTCOFs under N₂ atmosphere. **a** Co-TTCOF. **b** Ni-TTCOF. **c** H₂-TTCOF.

9. Does cobalt or Nickel leach into the solution after the run? Again this needs to be measured via ICP (post experiment measurement).

Response:

Thanks for your insightful comment. We have conducted the ICP tests of Co-TTCOF and Ni-TTCOF after long-time durability tests (> 40 h). Electrocatalysis was conducted (Co-TTCOF, -0.7 V (vs. RHE) and Ni-TTCOF, -0.9 V (vs. RHE)) after 40 h. After tests, negligible leaching of metal ions are detected for both Co or Ni in the electrolyte. This might be attributed to the strong covalent bond (C=N) form in the COF structures that can endow these materials with high durability.

Now we have added relative discussion in the revised manuscript (page 9, line 4).

10. Manuscript writing should be improved; the authors have failed to cite recent references in introduction part, In the introduction: ACS Catal. 2017, 7, 6120, ACS Appl. Mater. Interfaces, 2019, 11, 1520 and ACS Appl. Mater. Interfaces 2017, 9, 23843 are relevant under COF, CMP catalysis with metal porphyrin CMPs, COFs should be cited.

Response:

Thanks for your useful suggestion. We have carefully checked and improved the manuscript writing through the manuscript. Besides, we have properly cited relevant references in the revised manuscript and highlighted them in yellow (page 17, refs. 40,

41 and 38).

11. Electrocatalytic properties of COFs, which composed of free-base porphyrin, should be checked to verify the key role of Co^{2+} and Ni^{2+} as the active site.

Response:

Per suggestion, we have tested the electrocatalytic performances of H_2 -TTCOF (composed of free-base porphyrin). The gas chromatography (GC) analysis shows that H_2 is the primary reduction product over a wide potential range (-0.5 to -0.9 V vs. RHE), and only a small amount of CO is detected (FE_{CO} , 4.22%, -0.7 V). The performance is inferior to Co-TTCOF (FE_{CO} , 91.3%, -0.7 V) and Ni-TTCOF (FE_{CO} , 20.9%, -0.9 V) (Fig. 2c). This implies Co^{2+} or Ni^{2+} as the active sites in the porphyrin center indeed plays a key role during the reaction process.

Now we have added relative discussion in the revised manuscript (page 8, line 18).

Fig. 2c Electrocatalytic performances of M-TTCOFs. FE_{CO} is calculated over potential range from -0.5 to -0.9 V.

12. To established durability of your catalyst, you should mention material characterization data like HRTEM, FESEM, XPS after the CO_2 reduction experiment (post characterization data).

Response:

Thanks for your kind suggestion. According to your suggestion, we have added the HRTEM, FESEM, XPS tests of Co-TTCOF after electrocatalysis. To test it, the electrocatalyst is scraped from the surface of carbon cloth after electrocatalysis and characterized. The SEM and TEM images of Co-TTCOF agree well with the state before electrocatalysis, indicating Co-TTCOF can maintain its morphology after electrocatalysis (Supplementary Fig. 37). Specially, some inevitable acetylene black particles are detected on the surface of Co-TTCOF owing to the fabrication procedure of working electrode. Besides, the XPS tests after electrocatalysis show that the

valence state of Co(II) remains almost unchanged ($\text{Co}2p_{3/2}$, 780.86 eV and $\text{Co}2p_{1/2}$, 760.16 eV) when compared with that of Co-TTCOF ($\text{Co}2p_{3/2}$, 780.84 eV and $\text{Co}2p_{1/2}$, 796.14 eV) before electrocatalysis (Supplementary Fig. 38). These results indicate Co-TTCOF to be excellent electrocatalyst with high durability, which might be attributed to the strong covalent bonds generated in COFs.

We have now included relative discussion in the revised manuscript (page 8, line 46) and Supporting Information (page 37, Supplementary Fig. 37; page 38, Supplementary Fig. 38).

Supplementary Figure 37. SEM and TEM images of Co-TTCOF after long-time durability tests. **a** SEM image. **b** TEM image.

Supplementary Figure 38. High-resolution XPS spectrum of Co-TTCOF after long-time durability tests.

13. The efficacy of the catalyst is well represented by TON and this value should be given. What is TON for this process?

Response:

Thanks for your useful suggestion. Turnover number (TON) is defined as the mole of reduction product generated per electrocatalytic active site over a given period of time, which is an important parameter to evaluate the catalysis performance of electrocatalyst. We have calculated the TON of Co-TTCOF for this process. Notably, the TON (CO) of Co-TTCOF as high as 40142 in just 10 h and can reach up to 141479 after 40 h (Supplementary Fig. 36).

The TON for CO was calculated as follows:

$$\text{TON} = \frac{Q \times E_F}{N \times F \times n_{tot}}$$

where Q is the total charge passed in time, E_F is the Faradaic efficiency for the desired product, N is the number of electrons in the half reaction ($N = 2$ for CO_2 to CO conversion), F is the Faraday constant ($F = 96485 \text{ C mol}^{-1}$ electrons), and n_{tot} is the total moles of catalyst employed in the electrolysis. The TON is calculated on the basis of the actually catalytic activity.

We have now included the relative discussion in the revised manuscript (page 8, line 44) and Supporting Information (page 36, Supplementary Fig. 36).

Supplementary Figure 36. Plots of CO and H₂ evolving turnover number versus time for Co-TTCOF. As shown in the images, the TON (CO) is as high as 40142 in just 10 h and can reach up to 141479 after 40 h.

14. The NMR is too broad and unresolved. Deconvolution of the solid state ¹³C NMR spectra would be needed to state this COFs structure (e.g. Macromolecules, 2018, 51, 3088).

Response:

Thanks for your insightful suggestion. According to your suggestion, we have tested the solid state ¹³C NMR spectra of Co-TTCOF with modified test parameters (now we use 600 MHz (400 MHz in our previous test, which caused too broad peak) ¹³C NMR instrument (Bruker AVANCE III 600)) to improve the resolution. The

formation of C=N bond is proved by the existence of resonance signal around 157.3 ppm (peak a) (Supplementary Fig. 4a, b). The α -pyrrolic carbon present in the porphyrin moiety shows a peak at \sim 144.5 ppm (peak b1). The dithiole carbon present in the tetrathiafulvalene shows a peak at \sim 113.8 ppm (peak d1). The peak around \sim 115.8 ppm (peak d2) is attributed to β -pyrrolic carbon and methine carbon of the porphyrin macrocycle (Supplementary Fig. 4d). Besides, the peaks among the range from 124 to 136 ppm (peak c) can reflect the existence of phenyl moiety in Co-TTCOF.

We have properly cited relevant references (ref. 39) and added relative discussion in the revised manuscript (page 4, line 22) and Supporting Information (page 4, Supplementary Fig. 4).

Supplementary Figure 4. Representative ^{13}C cross polarization solid-state NMR spectra of Co-TTCOF. **a** The schematic structure of Co-TTCOF. **b** Solid state ^{13}C NMR spectrum of Co-TTCOF. **c-e** The spectral deconvolution of Co-TTCOF in **b**. The test frequency was 600 MHz in a ^{13}C NMR (Bruker AVANCE III 600).

Reviewer 2:

In this work, the authors reported a series of metalloporphyrin-tetrathiafulvalene based covalent organic frameworks (M-TTCOFs). Tetrathiafulvalene, serving as

electron donor or carrier, can construct oriented electron transmission pathway with metalloporphyrin. The obtained M-TTCOFs can be used as electrocatalysts with high FE_{CO} (91.3%, -0.7 V) and possess remarkable cycling stability (> 40 h) in electrocatalytic CO_2RR . After exfoliation, the FE_{CO} of COF nanosheets (~5 nm) is higher than 90% in a wide potential range from -0.6 to -0.9 V and the maximum FE_{CO} can reach up to almost 100% (99.7%, -0.8 V), which have been largely improved compared with bulk material. The electrocatalytic CO_2RR mechanisms are discussed and revealed by DFT calculations. The data presented can fully support the conclusion. This work is quite novel and important, and paves a new way in exploring porous crystalline materials in electrocatalytic CO_2RR . Therefore, I recommend the publication in Nature Communications after minor revisions.

I would like to suggest the followings to the authors:

1. What is the main concern of choosing tetrathiafulvalene as the desired electron donor to construct COFs? It would be nice to have some discussion in the introduction, from the viewpoint of structure design or target properties.

Response:

Thanks for your kind suggestion. In this work, we specially select tetrathiafulvalene (TTF) to construct metalloporphyrin-TTF based COFs. TTF is a sulfur-rich conjugated molecule with two reversible and easily accessible oxidation states (i.e., radical TTF^+ cation and TTF^{2+} dication) that have been widely studied as electron donor (*J. Am. Chem. Soc.* **2012**, 134, 12932). Co-porphyrin with inherent macrocycle conjugated π -electron system is very beneficial for electron mobility and Co(II) enables to be reduced to Co(I) during the process in many references (*Nat. Commun.* **2018**, 9, 4466; *Science* **2015**, 349, 1208). The connection of tetrathiafulvalene and metalloporphyrin will presumably create an oriented electron transmission pathway under the motivation of electric field and might facilitate the electron transfer efficiency to improve the CO_2RR property. As a proof-of-concept, thus-obtained metalloporphyrin-TTF based COFs exhibits excellent CO_2RR performances (e.g., Co-TTCOF, $FE_{CO}\%$ = 91.3, -0.7 V; long-time durability, 40 h).

We have now included relative discussion in the revised manuscript (page 3, line 3).

2. The detailed procedure about the test of the relatively direct current conductivity should be presented, such as pre-treatment of the samples and experimental operation. Besides, some discussion about this test should be provided.

Response:

Per suggestion, we have added the detailed procedures about the test of the relatively direct current conductivity in the “Characterizations and instruments” section in the revised manuscript (page 14, line 5). The detailed procedures are presented as follows: the relatively direct current conductivity tests were conducted with a probe station at room temperature (25 °C) under ambient conditions with a computer-controlled analogue-to-digital converter (2636B, Kethley). The conductive sample was pressed into a sheet using a ton of pressure in the mold, and the test voltage was among the range from -200 to 300 mV. Specially, H_2 -TTCOF presents

larger slope ($1/R$) value than COF-366-H₂ (constructed from 1,4-benzenedicarboxaldehyde and metalloporphyrin), which supports the structure superiority of the obtained M-TTTCOFs over other COF structures like COF-366-H₂ in electrocatalytic CO₂RR (Supplementary Fig. 24).

We have now included relative discussion in the revised manuscript (page 7, line 36).

Supplementary Figure 24. *I-V* profile tests and instrument. **a** Test instrument. **b** *I-V* profile of H₂-TTTCOF (red) and COF-366-H₂ (black).

3. In order to exclude the influence of substrate and additives, the carbon cloth with acetylene black and Nafion would be measured in CO₂-saturated 0.5 M KHCO₃ aqueous solution.

Response:

Thanks for your insightful comment. We have added the electrocatalytic performances of the carbon cloth with acetylene black and Nafion. The gas chromatography (GC) analysis shows that H₂ is the primary reduction product over a wide potential range (-0.5 to -0.9 V vs. RHE) (Supplementary Fig. 31). The results of this test indicate that the high CO₂RR performance indeed comes from M-TTTCOFs rather than substrate or additives.

We have now included relative discussion in the revised manuscript (page 8, line 15) and Supporting Information (page 31, Supplementary Fig. 31).

Supplementary Figure 31. FE_{CO} and FE_{H₂} of the carbon cloth with acetylene black and Nafion at different applied potentials in CO₂-saturated 0.5 M KHCO₃ aqueous solution.

4. Please provide the TON of Co-TTCOF for the production of H₂ and CO.

Response:

Thanks for your useful suggestion. Turnover number (TON) is defined as the mole of reduction product generated per electrocatalytic active site over a given period of time, which is an important parameter to evaluate the catalysis performance of electrocatalyst. We have calculated the TON of Co-TTCOF for this process. Notably, the TON (CO) of Co-TTCOF as high as 40142 in just 10 h and can reach up to 141479 after 40 h (Supplementary Fig. 36).

The TON for CO was calculated as follows:

$$\text{TON} = \frac{Q \times E_F}{N \times F \times n_{tot}}$$

where Q is the total charge passed in time, E_F is the Faradaic efficiency for the desired product, N is the number of electrons in the half reaction ($N = 2$ for CO₂ to CO conversion), F is the Faraday constant ($F = 96485 \text{ C mol}^{-1}$ electrons), and n_{tot} is the total moles of catalyst employed in the electrolysis. The TON is calculated on the basis of the actually catalytic activity.

We have now included relative discussion in the revised manuscript (page 8, line 44) and Supporting Information (page 36, Supplementary Fig. 36).

Supplementary Figure 36. Plots of CO and H₂ evolving turnover number versus time for Co-TTCOF. As shown in the images, the TON (CO) is as high as 40142 in just 10 h and can reach up to 141479 after 40 h.

5. The authors mentioned the performances of diverse electrocatalysts (e.g., metals, metal dichalcogenide and metal oxide) in the introduction. It would be better to list their CO₂RR performances in Supplementary file.

Response:

Thanks for your kind suggestion. According to your suggestion, we have added the performances of diverse electrocatalysts in the revised Supporting Information (page 44, Supplementary Table 2).

Supplementary Table 2. The summary of CO₂ electroreduction performances for reported electrocatalysts and M-TTCOFs.

Catalysts	Electrolyte	E (V vs. RHE)	Main product	FE (%)	Ref.
Co-TTCOF NSs	0.5 M KHCO ₃	-0.8 V	CO	99.7	This work
Co-TTCOF	0.5 M KHCO ₃	-0.7 V	CO	91.3	This work
Ni-TTCOF	0.5 M KHCO ₃	-0.9 V	CO	20.9	This work
H ₂ -TTCOF	0.5 M KHCO ₃	-0.7 V	CO	4.22	This work
COF-366-Co	0.5 M KHCO ₃	-0.55 V	CO	90.0	1

COF-366-F-Co	0.5 M KHCO ₃	-0.55 V	CO	87.0	2
COF-300-AR on Ag film	0.1 M KHCO ₃	-0.85 V	CO	80.0	3
Cu nanosheets	2 M KOH	NA	acetate	48.0	4
single-atom iron	0.5 M KHCO ₃	-0.47 V	CO	> 90	5
WSe ₂ 2D nanoflake	ionic liquid	-0.76 V	CO	> 80	6
Co ₃ O ₄ -CDots-C ₃ N ₄	0.5 M KHCO ₃	-0.6 V	CO	89	7
Cu-based nanoparticles	0.1 M KHCO ₃	-1.1 V	C ₂ H ₄	57.3	8
SnO ₂ nanosheets	0.5 M NaHCO ₃	-1.6 V (vs. Ag/AgCl)	HCOO ⁻	87	9

6. The exfoliated COF was produced by high-frequency sonication, while the specific work parameters of ultrasound such as frequency and temperature were not listed. Please add it.

Response:

Thanks for your kind suggestion. According to your suggestion, we have added the specific work parameters of ultrasound such as frequency (1000 W) and temperature (25 °C) in “Materials and Methods” section in the revised manuscript (page 13, line 27).

7. Metalloporphyrin or tetrathiafulvalene based COFs are a class of promising materials for electrocatalysis. Some important works of metalloporphyrinic or tetrathiafulvalenic COFs (e.g., *Chem. Sci.*, **2014** 5, 4693; *Chem. Eur. J.* **2014**, 20, 14614; *Angew. Chem. Int. Ed.* **2019**, 58, 6430, etc.) might be better cited to further support the novelty of the specially designed metalloporphyrin-tetrathiafulvalene based COFs.

Response:

Thanks for your kind suggestion. The related references have been properly cited in the revised manuscript (refs. 23, 35 and 36).

8. The description of the caption in the Supplementary Figure 23 and Supplementary Figure 25 are incorrect. For example, “Ar-saturated” in “the FE_{CO} and FE_{H₂} of pure carbon cloth at different applied potentials in Ar-saturated 0.5 M KHCO₃ aqueous solution” is “CO₂-saturated”.

Response:

Per suggestion, we have corrected the word “Ar-saturated” to “CO₂-saturated” and checked through the manuscript to avoid this problem (page 30, Supplementary Fig. 30; page 32, Supplementary Fig. 32).

REVIEWERS' COMMENTS:

Reviewer #1 (Remarks to the Author):

Ms. No.: NCOMMS-19-26978

Title: 'Efficient Electron Transmission in Covalent Organic Framework Nanosheets for Highly Active Electrocatalytic CO₂ Reduction'

Authors: Hong Jing Zhu, Meng Lu, Yi Rong Wang, Su-Juan Yao, Mi Zhang, Yu He Kan, Jiang Liu, Yifa Chen, Shun Li Li and Ya Qian Lan.

Comments:

The authors properly addressed all the concerns expressed in my previous review. Therefore, this reviewer would recommend publication as it is.

Reviewer #2 (Remarks to the Author):

The authors have addressed the major part of the concerns and the additional data are useful to make the manuscript much better. The quality of the manuscript is also much improved to meet the standard of Nature Communications and thus I recommend for the publication. There are some minor issues that need to improve before publication.

1. I would advise that the TON data for the H₂ of Co-TTCOF would be further discussed in the main text.

2. The corresponding formula like calculation detail for the percent of electrochemically active cobalt or nickel would be provided in the SI for easy reading.

Responses to the Reviewers' Comments

Reviewer 2:

The authors have addressed the major part of the concerns and the additional data are useful to make the manuscript much better. The quality of the manuscript is also much improved to meet the standard of Nature Communications and thus I recommend for the publication. There are some minor issues that need to improve before publication.

1. I would advise that the TON data for the H₂ of Co-TTCOF would be further discussed in the main text.

Response:

Thanks for your kind suggestion. The TON data for the H₂ of Co-TTCOF is discussed in the main text. The TON (H₂) of Co-TTCOF is 4014 in 10 h and can reach up to 14148 after 40 h.

We have now included the relative discussion in the revised manuscript (page 8, line 48) and Supporting Information (page 36, Supplementary Fig. 38).

2. The corresponding formula like calculation detail for the percent of electrochemically active cobalt or nickel would be provided in the SI for easy reading.

Response:

Per suggestion, we have added the detailed calculation in the revised manuscript (page 15, line 42).